# Cultural alignment of machine-vision representations

**Necdet Gürkan**
School of Business
Stevens Institute of Technology
Hoboken, NJ 07030
ngurkan@stevens.edu

**Jordan W. Suchow**
School of Business
Stevens Institute of Technology
Hoboken, NJ 07030
jws@stevens.edu

## Abstract

Deep neural network representations of visual entities have been used as inputs to computational models of human mental representations. Though these models have been increasingly successful in predicting behavioral and physiological responses, the implicit notion of "human" that they rely upon often glosses over individual-level differences in subjective beliefs, attitudes, and associations, as well as group-level cultural constructs. Here, we align machine-vision representations to the consensus among a group of respondents by extending Cultural Consensus Theory to include latent constructs structured as fine-tuned pretrained machine-vision systems. We apply the model to a large-scale dataset of people's first impressions of others. We show that our method creates a robust mapping between machine-vision representations and culturally constructed human representations.

## 1 Introduction

All areas of life are affected by inferences drawn from visual information, including perceptions of whether a political candidate is competent (Ballew & Todorov, 2007), a food product is nutritious (Mai et al., 2016), a clothing style projects intelligence (Rosenbusch et al., 2021), or a brand is appealing (Luffarelli et al., 2019). Such inferences determine the social, occupational, consumption, and health outcomes of individuals, as well as the political and economic makeup of society (Bhatia & Aka, 2022). It is thus of critical significance to social and behavioral scientists to develop predictive and explanatory models of people's psychological inferences about entities from visual information.

Deep learning methods, which learn expressive feature representations of high-dimensional images, have been used in models of human representations for perceptual tasks, such as predicting the memorability of objects in images (Khosla et al., 2015) and typicality judgments (Lake et al., 2015). Cognitive scientists have combined representations obtained from machines with psychological models to examine the correspondence between two (Peterson et al., 2016) and bring them into closer alignment (Roads & Love, 2021; Battleday et al., 2020; Sanders & Nosofsky, 2020).

Although machine-vision models are increasingly successful in predicting behavioral and physiological responses of humans, the implicit notion of "human" that they rely upon often glosses over individual-level differences in subjective beliefs, attitudes, and associations, as well as group-level cultural constructs. For example, one subculture may perceive face tattoos as taboo, whereas another may find them to be stylish and meaningful expressions of individual or group identity. Consequently, different subsets of the population may have quite different representations of faces having tattoos when using them to infer additional psychological attributes that might also be influenced by their beliefs, associations, and attitudes.

Here, we align machine-vision representations to the consensus among a group of respondents by extending Cultural Consensus Theory to include latent constructs structured as fine-tuned pretrained

4th Workshop on Shared Visual Representations in Human and Machine Visual Intelligence (SVRHM) at the Neural Information Processing Systems (NeurIPS) conference 2022. New Orleans.

machine-vision systems. Cultural Consensus Theory (CCT) is a model-based statistical technique to estimate the consensus among a group of respondents when the ground truth is unavailable or undefined (e.g., in collective memories of events, beauty norms, or social relationships in covert networks). CCT offers a practical basis to capture individual and group differences using participant-level data. However, it has several limitations to aligning learned deep-network representations with human psychological inferences, such as ignoring the correlation between items and creating a mapping from intermediate representation to human inferences. In this paper, we extend CCT by enabling culturally held beliefs to form a latent regression construct that maps from a question to a consensus answer through an intermediate representation. Our approach aligns learned representations of a group and individual level and is thus able to capture differences in psychological processes and behaviors across individuals and groups.

The plan of the paper is as follows. First, we give a brief introduction to CCT and its applications. Next, we introduce the CCT model for the continuous response, which serves as a base model for our extension. Then, we describe our extension of the CCT. Finally, we present and discuss the results of fitting the model to human psychological inferences of first impressions of faces.

## 2    Cultural consensus theory

Since its introduction in *American Anthropologist* (Romney et al., 1986), CCT has been used in many applications. The framework can be used to infer what cultural beliefs influence social practices and the degree to which individuals know or show those beliefs. A key feature of CCT is that the researcher does not assume to know the culturally "correct answers" to questions a priori. Instead, CCT treats these answers as latent variables, which are estimated from the respondents' observed data. Estimated consensus knowledge does not correspond to ground truth knowledge. The goal of a CCT is to find if the respondents share an underlying consensus, regardless of whether that consensus corresponds to some exogenously defined objective truth (Batchelder et al., 2018). The latent truth assumption sets CCT apart from the other psychometric test theories (Baker, 2001; De Boeck & Wilson, 2004; Fox, 2010), where the goal is to measure the respondents' ability in well-defined objective truth. When applied, CCT provides an assignment of respondents to cultures, a consensus answer for each question for each culture, a measure of each respondent's cultural alignment, a difficulty level for each question, and, in its continuous variant, a shift and scale bias for each respondent.

The CCT framework has been applied to find a practical and concise definition of beliefs that are accepted by a group who shares common knowledge. These models have been widely used to study mental health (Alang, 2018), cognitive evaluation (Heshmati et al., 2019), feelings of love (Oravecz & Vandekerckhove, 2020), organizational culture (Rinne & Fairweather, 2012), and norm formation in online communities (Gurkan & Suchow, 2022). In addition, the features of CCT are well-suited for "wisdom of crowd" application in which each respondent in a group provides a single prediction regarding to an event that has a known correct value (Prelec et al., 2017; de Courson et al., 2021).

### 2.1    Continuous cultural consensus theory

As a starting point, consider the Continuous Response Model (CRM) (Anders et al., 2014), a cultural consensus model for continuous data introduced by observations of the random response profile matrix $\mathbf{X}_{ik} = (X_{ik})_{N \times M}$ for $N$ respondents and $M$ items, where each respondent's response falls within $(0, 1)$ or a finite range that permits a linear transformation to $(0, 1)$. The CRM links the random response variables in $(0, 1)$ to the real line with the logit transform, $X^* = \text{logit}(X_{ik})$. Therefore, each item also has a consensus value in $(-\infty, \infty)$.

The CRM is formalized and further explained by the following axioms:

**Axiom 1** (*Cultural truths*). There is a collection of of $V \geq 1$ latent cultural truths, $\{T_1, ..., T_v, ..., T_V\}$, where $T_V \in \prod_{k=1}^{M}(-\infty, \infty)$ . Each participant $i$ responds according to only one cultural truth (set of consensus locations), as $T_{\Omega_i}$, where $\Omega_i \in \{1, ...., V\}$, and parameter $\Omega = (\Omega_i)_{1 \times N}$ denotes the cultural membership for each informant.

**Axiom 2** (*Latent Appraisals*). It is assumed that each participant draws a latent appraisal, $Y_{ik}$, of each $T_{\Omega_i k}$, in which $Y_{ik} = T_{\Omega_i k} + \epsilon_{ik}$, The $\epsilon_{ik}$ error variables are distributed normal with mean 0 and standard deviation $\sigma_{ik}$.

**Axiom 3** (*Conditional Independence*). The $\epsilon_{ik}$ are mutually stochastically independent.

**Axiom 4** ( *Precision.*). There are knowledge competency parameters $\mathbf{E} = (E_i)_{1 \times N}$ with all $E_i > 0$, and item difficulty parameters specific to each cultural truth $\Lambda = (\lambda_k)_{1 \times M}$, $\lambda_k > 0$ such that

$$\sigma_{ik} = \lambda_k / E_i. \tag{1}$$

If all item difficulties are equal, then each $\lambda_k$ is set to 1.

**Axiom 5** (*Response Bias*). There are two respondent bias parameters that act on each respondent's latent appraisals, $Y_{ik}$, to arrive at the observed responses, the $X_{ik}$. These include a scaling bias, $\mathbf{A} = (a_i)_{1 \times N}, a_i > 0$; and shifting bias $\mathbf{B} = (b_i)_{1 \times N}, -\infty < b_i < \infty$, where

$$X_{ik}^* = a_i Y_{ik} + b_i. \tag{2}$$

These axioms are developed to model the continuous response of respondents that differ in cultural competency, $E_i$, and response biases, $a_i$ and $b_i$, to items that have different shared latent truth values. The respondents have a latent appraisal of these item values with a mean at the item's consensus location plus some error, which depends on their competence level and the item difficulty. Axiom 1 locates the item truth values in the continuum. Axiom 2 defines the appraisal error is normally distributed with mean zero. Axiom 3 sets the appraisals are conditionally independent given the respondents' cultural truth and the error standard deviations. Axiom 4 specifies the standard appraisal error that depends on the respondent's competence and item difficulty. Axiom 5 covers each respondent's response bias and location response tendencies on the scale.

## 3 Extending CCT with infinite deep neural network latent constructs

CCT operationalizes the structure of culturally held beliefs as what amounts to a lookup table, one with keys that are questions and values that are answers. The questions and answers themselves have no defined internal structure and are not linked to each other in any way except through correlations across respondents' answers. However, such a formulation has several limitations. First, because it treats each question/answer pair as an island entire of itself, information gleaned from one question does not inform our understanding of other questions. Second, for the same reason, the number of questions that must be tested to characterize a culture scales linearly with the number of culturally held beliefs. And third, there is no way to leverage insights from existing knowledge bases that provide structured information about known entities and their relations. Here, we propose to extend cultural consensus theory by enabling culturally held beliefs to take the form of mathematical objects more complex than a lookup table. We operationalize culturally held beliefs as an algorithmic latent construct, a function that maps from a question to a consensus answer through an intermediate representation. In particular, we consider an intermediate representation that has the structure of a pretrained deep neural network that is fine-tuned with a linear readout layer specific to the culture.

Extending CCT in this way enables us to create a mapping between deep features, $\phi_k$, and the cultural consensus, $T_{vk}$. To fit the latent construct, we introduce a latent variable, $\omega_{\Omega_i}$, that represents the regression weights for each participant's cultural membership, $\Omega_i$. The relation between the weights of latent construct, $\omega_{\Omega_i}$, and high-dimensional expressive machine features, $\phi_k$, is given by the regression equation

$$\begin{aligned} T_{vk} &= \phi_k \omega_{\Omega_i}^T \\ Y_{ik} &= T_{vk} + \epsilon_{ik}, \end{aligned} \tag{3}$$

where $\epsilon_{ik}$ is the error variables in Axiom 4 (Eq. 1). We replace the consensus location described in Axiom 1 with a function that takes as input machine features and corresponding weights for each feature.

In this work, we use Bayesian Ridge regression to regularize parameters in the estimation procedure. The prior for the coefficients, $\omega_{\Omega_i}$, is given by a spherical Gaussian:

$$p(\omega_{\Omega_i} \mid \zeta) = \text{Normal}(\omega_{\Omega_i} \mid 0, \zeta^{-1} \mathbf{I}_p), \tag{4}$$

with the prior over $\zeta$ assumed to be Gamma distributed, the conjugate prior for the precision of the Gaussian.

We further extended the CCT model to enable an unbounded number of cultures using a Dirichlet Process prior (via a stick-breaking process) over the cluster assignments (see *Appendix*). This modification produces a Bayesian nonparametric model, where the number of instantiated cultures grows with the complexity of the observed data. As opposed to the original CCT framework, which uses a fixed number of cultures tuned by the experimenter, the Bayesian nonparametric variant of the model provides a posterior over the entire space of partitions.

## 3.1 Hierarchical specification of the extended CCT

In this section, we demonstrate our extended CCT hierarchically such that the item and participant traits are each considered to be samples from population (or the cultural) distributions. We use a Bayesian hierarhical model that allows for multiple processes to contribute to a single set of observed data (Lee, 2011). In hierarchical modeling, population distributions are specified for the regular parameters using hyperparameters. These hyperparameters are estimated from their own distributions and can represent the central tendency within each trait across items or participants, which may be unique to each dataset. The hierarchical structure of our generative model of human subjective inferences of faces is described below with a following explanation.

$$\omega_{\Omega_i} \sim \text{Normal}(0, \zeta^{-1}) \qquad \text{Coefficient weights}$$
$$T_{vk} = \phi_k \omega_{\Omega_i}^T + \epsilon_{ik} \qquad \text{Latent construct item location}$$
$$\lambda_{vk} \sim \text{Normal}(\mu_{\lambda_v}, \tau_{\lambda_v}) \qquad \text{Item difficulty}$$
$$\log(E_i) \sim \text{Gamma}(\alpha_{E_{\Omega_i}}, \kappa_{E_{\Omega_i}}) \qquad \text{Participant Competency}$$
$$a_i \sim \text{Normal}(\mu_{a_{\Omega_i}}, \tau_{a_{\Omega_i}}) \qquad \text{Informant scaling bias}$$
$$b_i \sim \text{Normal}(\mu_{b_{\Omega_i}}, \tau_{b_{\Omega_i}}) \qquad \text{Informant shifting bias}$$
$$\Omega_i \sim \text{Categorical}(\pi) \qquad \text{Group membership}$$
$$\pi \sim \text{stickbreaking}(\beta) \qquad \text{Probability of group membership}$$

We used Bayesian Ridge Regression as a latent construct to estimate item consensus, $T_{vk}$, which are located on the real line, paramaterized with a mean and precision (inverse variance). The location of the item consensus is calculated by taking the dot product of two row matrices. The other model parameters, which are each located on the positive half-line: $\lambda_i$, $b_i$, and $a_i$, are assumed to be sampled from a normal population distribution. $E_i$ is log-transformed and sampled from gamma distribution. Note that the three primary informant parameters remain singly-indexed by *i*, through a technique in which their distribution is specified by their group membership $\Omega_i$. The probability of being in any give group is assigned a stick-breaking prior, and this allows for varying probabilities of being in any given group, of *V* groups; note that *V* is unknown priori and needs to be estimated from observed data. We can remove the $\Omega_i$, and $\pi$ variables for single-truth estimation of our model. Prior distributions are set for a number of hyperparameters, while others are set to fixed values. The priors over hyperparamater distribution settings used for the model and stickbreaking process can be found in the next subsection.

## 4 Method

### 4.1 Data

The latent-construct CCT model is applied here to a large dataset of people's first impressions of faces Peterson et al. (2022). The dataset contains over 1 million judgments of 34 trait inferences for 1,000 face images. Each face is rated by 30 unique participant for each trait. In this dataset, the face images are generated using a synthetic photorealistic image generator, StyleGAN2 (Karras et al., 2020). The generator network component of StyleGAN2 models the distribution of face images conditioned on a 512-dimensional, unit-variance, multivariate normal latent variable. Thus, a 512-dimensional representation is used for our modeling, associating latent this feature vector with a face through Ridge regression. We used 80% of the data for training and the remaining for validation.

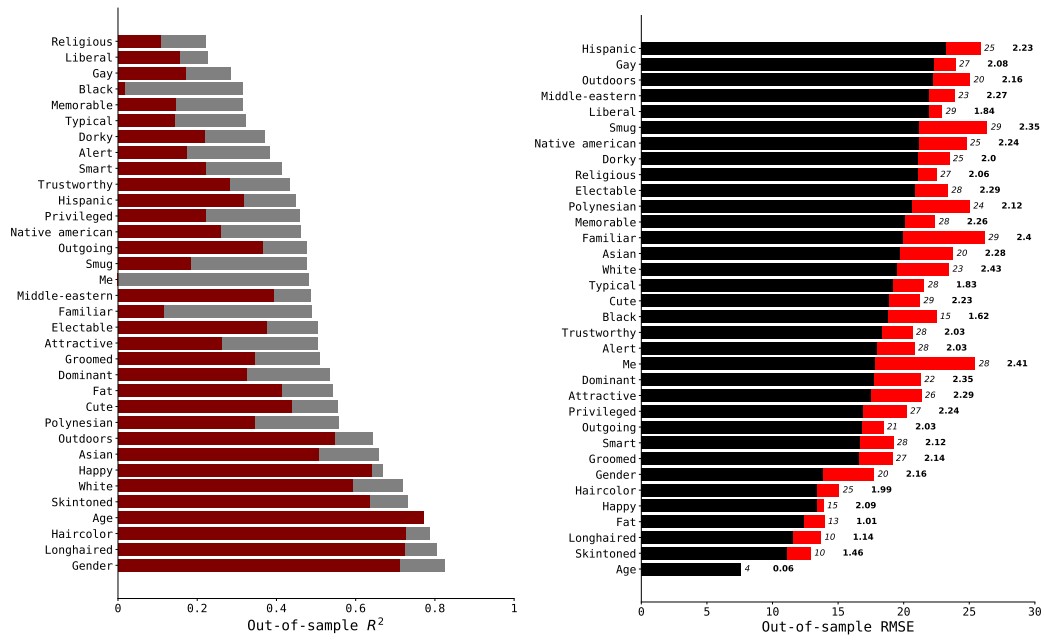

Figure 1: **Left:** Performance ($R^2$) for the extended single-truth (red bars) and multiple-truths (black bars) CCT models. **Right:** Prediction accuracy (RMSE) using the extended single-truth (red bars) and multiple-truths (black bars) CCT models. Italicized numbers next to the bars represent the number of cultures, and bold numbers indicates the entropy of the cluster-assignment distribution.

## 4.2 Implementation

The model was implemented in NumPyro (Phan et al., 2019) with the JAX backend (Bradbury et al., 2018). The model components were integrated into a single likelihood function and a set of prior distributions, needed to infer a posterior over the unobserved variables in our model using Discrete Gibbs and the NoU-Turn Sampler (NUTS) (Hoffman et al., 2014), a standard Markov chain Monte Carlo sampling algorithm, as implemented in Numpyro. We used 1 chain with 1,000 warm up samples and 1,000 draw samples, thereby obtaining 1,000 posterior samples for each experiment. We ensured that the posterior had converged by ensuring there were not divergence transitions.

## 5 Results

In this section, we demonstrate the alignment accuracy of our extended CCT to human psychological inferences of faces. To determine whether the participant-level data includes one vs. multiple consensuses, we fit a single-truth version of the extended CCT, which assumes that there is only one consensus for a given item, and a multiple-culture version of the extended CCT as described above. As shown in Fig. 1, the model accuracy is improved for each psychological attribute when we create a culturally specific mapping between learned machine-vision features and consensus beliefs. The model improvement is greater for the traits that have a high variation among participants because the single-truth model ignores such variation by assuming that a group of individuals do not share first impressions. For example, the single-truth model of *looks like you* is considerably outperformed by the multiple-truths model, which demonstrates better performance because it can cluster participants with similar beliefs about who looks like them into a shared culture and then fine-tune the latent construct to make more accurate predictions for that culture.

Fig. 1 (right) also reports the best-fit number of instantiated cultures, which varies considerably across traits. While the number of cultures indicate the number of consensuses in the data, it does not inform us about the uniformity of the cultural assignment distribution. An entropy-based metric addresses this problem by estimating the uncertainty of the cultural allocation of an unknown randomly chosen data point given a particular distribution of culture assignments. In Fig. 1, the smaller value of

entropy shows that there is a few large clusters and the larger the values of entropy are associated with more evenly distributed cultures.

## 6    Discussion

In this paper, we extended the CCT by enabling cultural held beliefs to take the form of a latent construct that maps from a psychological attribute to a consensus answer through an intermediate representation: a pretrained machine-vision system fine-tuned using Bayesian ridge regression. We then demonstrated that modeling individual and group consensus differences improves the alignment of machine representations with human psychological inferences of faces.

We note that our model is not restricted to human face-trait inferences. Integrating features extracted from deep convolutions neural networks into the CCT provides a unique opportunity to estimate a cultural consensus from any kind of visual entity for which a pretrained network is available. Also, machine-learning techniques have made it possible to extract rich quantitative representations from text and data from other modalities. These vector-space representations can also be aligned with human psychological inferences using the extended CCT model to form models of consensus beliefs in other domains.

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

# 7 Appendix

## 7.1 Extension: Infinite cultural consensus theory

CCT has been used to infer the cultural beliefs that influence social practices and the degree to which individuals understand and exhibit those beliefs (Romney et al., 1986). When there are multiple cultures, CCT analyzes eigenvalues obtained from the cross-participant correlation matrix to determine the number of cultures present. Two key problems with this approach are that it assumes a finite-dimensional representation that correctly characterizes the features of observed data and that there are few missing values in the observed data. Here, we introduce a Bayesian non-parametric model, where cultures are drawn from a Dirichlet Process (DP) by a way of the stick-breaking construction. This method provides a completely probabilistic approach to inferring the existing numbers of cultures that account to generate human face inference using CCT.

In non-parametric approach, the goal is to learn from the data without making any strong assumption about the class of distributions that might describe the data. The reason behind the method is that generative process for the data is improbable to belong any finite-dimensional distribution family, so it avoids the possible wrong assumption. The DP provides solutions to Bayesian non-parametric inferences by using prior over discrete distributions. This method is widely used in Bayesian nonparametric models as a infinite discrete priors for mixture models (Griffiths et al., 2003; Neal, 2000) and has been applied to psychometric models as a prior over probability Duncan & MacEachern (2008); Miyazaki & Hoshino (2009).

The DP was first formally introduced by Ferguson (1973). Consider $\mathcal{G}_0$ be a distribution over $\Theta$ and $\alpha$ be a positive real number. Then for any finite measurable partition $T_1, \ldots, T_r$ of $\Theta$ the vector $(\mathcal{G}(T_1, ..., \mathcal{G}(T_r))$ is random since $\mathcal{G}$ is random. We can state $\mathcal{G}$ is DP distributed with base distribution $\mathcal{G}_0$ and concentration parameter $\alpha$, written $\mathcal{G} \sim DP(\alpha, \mathcal{G}_0)$, if

$$(\mathcal{G}(T_1, ..., \mathcal{G}(T_r)) \sim \text{Dir}(\alpha\mathcal{G}_0(T_1), ..., \alpha(\mathcal{G}_0(T_r)) \tag{5}$$

for any finite measurable partition $T_1, ..., T_r$ of $\Theta$. The parameters $\mathcal{G}_0$ and $\alpha$ play intuitive roles in DP. The base distribution is the mean of DP for any measurable $T \subset \Theta$. The concentration parameter can be understood as an inverse variance. The higher $\alpha$ means that the DP will concentrate more of mass around the mean, which results in low variance. Random distributions, $\mathcal{G}$, drawn from the DP are discrete. We start by noting that discrete distributions can be written

$$\mathcal{G} = \sum_{r=1}^{\infty} w_r \delta_{\mathcal{Q}_r} \tag{6}$$

In this formulation, $w_r$ denotes the mixture weights assigned to the r[th] partition and $\delta_{\mathcal{Q}_r}$ represents a point mass distribution located at $\mathcal{Q}_r$. Although we assume an unlimited number of groups, each finite set of subjects contains representatives from a finite subset of those groups. This model is psychologically valid for CCT because people can behave in infinite number of ways, but only finite set of behaviors will be observed. With an infinite number of cultures, there is always the possibility that new subjects will exhibit unprecedented behavior (Navarro et al., 2006).

The simplest description of the DP applies the notion of a stick-breaking prior Ishwaran & Zarepour (2002). Sethuraman Sethuraman (1994) formalized a more constructive definition based on the stick-breaking representation. This representation can be described as following: Consider a stick with unit length. We then break of $\beta_1 \sim Beta(1, \alpha)$ of the stick and take the length of one of the pieces to be the first weight. For remaining pieces, we continue with this process for a countably infinite number of breaks. We then broke the remaining piece in two, using one of the resulting pieces as the second weight. This process continues for a countably infinite number of breaks. It results in an infinite set of stick-lengths, weights, that sum to 1 with probability 1

$$\beta_r \sim \text{Beta}(1, \alpha) \tag{7}$$

$$w_r = \beta_r \prod_{k=1}^{r-1} (1 - \beta_k) \quad r = 1, 2, 3, \ldots \tag{8}$$

Now, we can formulate a random distribution using the Eq. (6), where we take an infinite number of samples from a base distribution $\mathcal{G}_0$ and draw the weights as in Eq. (8).

We can then create an infinite number of cultural truths (consensus) that are estimated by individual level parameters in CCT framework. Respondents are assigned to the piece that explains their observed data. Their individual level parameters are estimated according to the piece (culture) they are assigned to. These parameters can be marginalized over the discrete cultural membership indicators $z$ by using an efficient posterior inference algorithm (e.g., ADI, NUTS, HMC) for learning the joint posterior of the remaining model parameters.

## 7.2 The priors over hyperparameter distributions

$$
\begin{aligned}
\zeta_{T_v} &\sim \text{Gamma}(10, 1) & \alpha &\sim \text{Gamma}(1, 10) \\
\mu_{\lambda_v} &= 1 & \tau_{\lambda_v} &\sim \text{Gamma}(1, 10) \\
\alpha_{E_{\Omega_i}} &\sim \text{Exponential}(0.1) & \kappa_{E_{\Omega_i}} &\sim \text{Exponential}(1) \\
\mu_{a_{\Omega_i}} &= 0 & \tau_{a_{\Omega_i}} &\sim \text{Gamma}(1, 10) \\
\mu_{b_{\Omega_i}} &= 0 & \tau_{b_{\Omega_i}} &\sim \text{Gamma}(1, 10)
\end{aligned}
$$

These prior settings are set for this particular applications. $\alpha$ is a concentration parameter for Beta distribution described in Infinite CCT section. $\alpha$ influences the number of clusters; large $\alpha$, greater number of clusters.

