# OpenReview forum: "Cultural alignment of machine-vision representations"
_NeurIPS.cc/2022/Workshop/SVRHM — SVRHM Poster_

### Official Review · Reviewer_ddYa · 2022-10-14
**Interesting work, would love to see it it at SVRHM 2022.**

**Rating:** 8
**Confidence:** 5

**Review:**

The paper is clear and easy to follow. The work would be of interest to the audience of SVRHM. Studying cultural consensus via intermediate latent representations is quite an original and fascinating direction.

A few details that I would have liked to have gotten in the paper are:
1. What is the pretrained deep network that was fine-tuned for this work?
2. What is the dimensionality of the Linear layer used for the fine-tuning?
3. Why did authors use Ridge regression as opposed to e.g. train with plain SGD etc.?

It would be nice to include this information in the main paper to facilitate reproducing results by other researchers. Would also like to hear what follow up directions this work opens for the field.

---

> ### Author Response · Authors · 2022-11-24
> **Respond to reviewer 4**
>
> Thank you for your useful comments on our work.
>
> We agree that the bullet points you mentioned need to be addressed to make the paper clearer and understandable. We will make sure we address the each point in our full-length paper.
>
> Thank you.

---

### Official Review · Reviewer_BJLu · 2022-10-14
**Review of "Cultural alignment of machine-vision representations"**

**Rating:** 6
**Confidence:** 3

**Review:**

This paper addresses an interesting problem of how latent space features used for image generation using deep generative models can also be informative of the sentiment that humans attribute to those images.

The upsides to this this paper are that instead of simply regressing to human judgements, which has been done by a quite a few cited papers such as Peterson et al. (2022), this paper considers how different cultures may align by taking CCT into account and evaluating both a single-truth and multi-truth model for each attribute. This is particularly interesting since people of different background may give different ratings, and this taken their respective cultures into account. Also, the metrics by which the authors evaluate their ridge regression models (R-squared and RMSE) are quite appropriate to assess how well the model fits the data.

The downsides here are that the authors claim they are aligning machine representations to human judgments, but the input to the regression model is simply the conditional input to StyleGAN2. What is the significance of regressing samples drawn from a multivariate normal distribution to human judgments? Instead, it would have made more sense for an intermediate representation to be used in the regression, as those would be machine representations, and not just samples from a random probability distribution.

On aesthetics, I strongly felt that the paper could have been written in a more clear an informative way, and could use more figures.

---

> ### Author Response · Authors · 2022-11-24
> **Respond to reviewer 3**
>
> Thank you for your feedback and comments.
>
> Here, we encoded each face to 512 dimensional StyleGAN space that is used to in our Ridge regression to generate the consensus for each item. We will further explore your suggestions and incorporate them in our future work.
>
> Thank you.

---

### Official Review · Reviewer_c7in · 2022-10-16
**An Interesting Way to Align Machine Vision Representations using a Psychometric Model**

**Rating:** 6
**Confidence:** 3

**Review:**

## Summary

This work uses the Cultural Consensus Theory (CCT), a Psychometric model that estimates the consensus of a group of participants as a latent variable that shapes their judgements, to finetune DNN visual representations and show that it can improve performance in predicting human subjective judgements on face stimuli (e.g. smug, memorable, familiar, etc). They show that, for some of these judgements, being able to account for multiple cultures can boost predictions of human judgements over assuming only one consensus.

## Strengths
* The work is well motivated, because being able to finetune these models to be able to account for multiple possible consensuses is an important direction to aligning human-machine visual representations.

* The authors do a really cool extension using Bayesian nonparametrics to make CCT compatible with an unbounded number of cultures/consensuses.

## Suggestions

* I think the results can be shown with a little more clarity. For instance, in Fig 1. right, RMSE is not defined. I assume it could mean root mean squared errors, but then all the black bars are considerably larger than the red bars, suggesting that the multiple truths models have higher error than the single truth models, which goes against what is said in the discussion?

* For at least Fig 1. left, I was surprised by how many traits for which the single-truth models still do better than multiple-truths. For example, for cute, red bar is still considerably higher than grey bar, though maybe I might be reading these wrong (see first suggestion). I'm wondering if the dataset's participant bias (primarily white and in the U.S) has something to do with there being a lower number of consensuses. I think collecting/using a dataset that draws from a much more diverse participant pool would allow this method to shine the most.

---

> ### Author Response · Authors · 2022-11-24
> **Respond to reviewer 2**
>
> Thank you for your beneficial comments on our work.
>
> RMSE (Root Mean Square Error) bars show the predictive power of our model. We compared our single vs multiple culture CCT model to investigate whether latent cultural construct exists in the first impressions of faces. Therefore, lower RMSE score means that the model is doing better job (having lower error score) on unseen items. Similarly, the left sub-plot in Fig1. shows the R-squared values on out-of-sample items (higher R-squared means the model is more generalizable on unseen samples). Age is an only trait that our single and multiple culture CCT model performed identically which is expected.
>
> We will pay more attention to explain the results in our future work. Thank you.

---

### Official Review · Reviewer_ZhLL · 2022-10-16
**Cultural alignment of machine-vision representations**

**Rating:** 6
**Confidence:** 4

**Review:**


Strengths
- This work proposes a novel extension of Cultural Consensus Theory (CCT).
- The new CCT framework was applied to a dataset from Peterson et al. 2016, addressing an important limitation of this prior work, considering responses made by different groups of people rather than in aggregate.

Weaknesses/improvements
- An ethical implications section should be added to this work. A good model for this would be the ethical implications section of Peterson et al. 2016 as this work builds on their data and has similar potentials for miss-use. There is extra space on page 5 to fit this in.
- Further analysis of the size and spread of the clusters found could be an interesting addition to this work as it could aid in collecting more diverse datasets and understanding current dataset biases.

Misc
- line 132 extra word, "model that were"
- line 175 extra word, "for from"

---

> ### Author Response · Authors · 2022-11-24
> **Respond to reviewer 1**
>
> Thank you for your useful comments to further develop this study.
>
> We agree that an ethical implications of this paper would be very useful to identify the human biases and reduce them by creating more diverse dataset or training humans accordingly. We are aiming to include them in the journal submission version of this work.
>
> Again, thank you!